# Microwave-Triggered 4D Automatic Color Change in 3D-Printed Food Materials Incorporating Natural Pigments

**DOI:** 10.3390/foods12102055

**Published:** 2023-05-19

**Authors:** Yaolei Zhu, Zhenbin Liu, Xiaofan Zhang, Chaojun He, Xinxin Zhang

**Affiliations:** 1College of Food and Drug, Luoyang Normal University, Luoyang 471934, China; 2School of Food Science and Engineering, Shaanxi University of Science and Technology, Xi’an 710021, China; 3School of Food Science and Technology, Jiangnan University, Wuxi 214122, China

**Keywords:** 4D printing, color change, microwave heating, 3D printing

## Abstract

The feasibility of using microwaves to quickly stimulate automatic color change in 3D-printed food containing curcumin or anthocyanins was studied. Firstly, with a dual-nozzle 3D printer, stacked structures included mashed potatoes (MPs, upper part, containing anthocyanins) and lemon juice–starch gel (LJSG, lower part) were 3D-printed and post-treated using a microwave. The results indicated that the viscosity and gel strength (indicated by the elastic modulus (G′) and complex modulus (G*)) of LJSG were improved with the increase in starch concentration, while water mobility was reduced. During microwave post-treatment, the color change speed was negatively correlated with the gel strength but positively correlated with the diffusion of H^+^ and anthocyanin concentration. Secondly, nested structures were 3D-printed using MPs containing curcumin emulsion and baking soda (NaHCO_3_). During microwave post-treatment, the curcumin emulsion structure was destroyed, and NaHCO_3_ was decomposed, along with an increase in alkalinity; thus, the automatic color change was achieved with the automated presentation of hidden information. This study suggests that 4D printing could enable the creation of colorful and attractive food structures using a household microwave oven, leading to more imaginative solutions regarding personalized foods, which may be particularly important to people with poor appetites.

## 1. Introduction

The technology of 3D printing has swiftly emerged as an alternative method of food processing owing to its numerous advantages. One of its primary benefits is the capacity to create customized food designs, enabling personalized nutrition for consumers. Additionally, it simplifies the food supply chain and broadens the range of available food materials, according to a recent study [1,2].

The methods using 4D printing technology involve predictable automatic changes in samples after 3D printing under certain specific stimuli, such as pH, temperature, and dehydration [3,4]. Automatic controllable changes in the color and shape of samples after 3D printing can be classified as 4D printing. Anthocyanins and curcumin are natural pigments that have different molecular structures and exhibit different colors at different pH levels [5]. They might be potential stimulus response materials to be used in 4D printing methods related to changes in color. However, the number of studies available in the literature for color-changing 4D food printing is limited. Ghazal et al. (2019) [6] and He et al. [7] utilized the principle of color change in anthocyanins at different pH levels, by spraying acidic solutions on the surface of printed samples or printing materials containing anthocyanins with different acid–base materials, but in the above studies, significant changes in color often required several hours or even longer, which hinders its application. It is highly required that a rapid color change in 3D-printed foods be achieved to realize the broader implication of this method. Microwaves are high-frequency electromagnetic waves that can penetrate materials and cause polar or dipolar molecules to rapidly change direction and rapidly increase the temperature, compared with conventional conductive heating [8]. Considering the quick heating process of microwaves and that microwave ovens are popular household appliances, a quick 4D printing process triggered by microwave treatment is desirable. To the best of our knowledge, there are few publications that utilize microwaves as a heating source to achieve a quick automatic change in the color of 3D-printed structures. Guo et al. [9] used a microwave as the stimulus source and gelatin–gum Arabic complex coacervates as the stimulus material to realize changes in the color and flavor of the printed dough. Chen et al. [10] used a microwave to stimulate color change in curcumin lotus root gel printed using 3D technology. However, in their studies, only one kind of food material was used to construct food structures, which restricts the development of multi-ingredient and multi-color foods. In our study, using dual-nozzle 3D printing technology, food structures containing both responsive parts (with pH-sensitive plant pigments) and stimulating parts (with acid or alkali) were constructed to achieve automatic color changes during post-processing. To realize the production of highly attractive multi-material constructs with higher geometric complexity and appealing appearance, a dual-nozzle printer was used in this study to create 4D-printed color-changing foods.

This manuscript involves an investigation of automatic color change driven by pH changes in 3D-printed food containing anthocyanins and curcumin during post-processing with microwave heating. First, a stacked structure was constructed using anthocyanin-containing mashed potatoes (MPs) and lemon juice–starch gel (LJSG), and the relationship between water mobility, rheological properties, and color-changing speed was studied. Then, based on the MPs, a nested structure containing curcumin emulsion and baking soda (NaHCO_3_) was constructed, and the automatic presentation of hidden information based on color changes was achieved during microwave post-processing. This study can provide a basis for using 3D printing technology to produce personalized 4D-printed color-changing food and is conducive to further leveraging the personalized customization advantages of 3D printing technology.

## 2. Materials and Methods

### 2.1. Materials

Potato flakes were purchased from Shijiazhuang Lingfeng Agricultural Products Development Co., Ltd. (Shijiazhuang, China). Xanthan gum and k-carrageenan gum were purchased from Shandong Yosolf Chemical Technology Co., Ltd. (Jinan, China). Concentrated lemon juice was provided by Guangdong Jiahao Food Co., Ltd. (Zhongshan, China). Potato starch was purchased from Shanghai Tianyu Food Co., Ltd. Blueberry anthocyanin (25% content) was purchased from Xi’an Shengqing Biotechnology Co., Ltd. (Xi’an, China). Curcumin (95% content) was purchased from Hebei Tianxu Biotechnology Company (Handan, China). Food-grade Tween 80 was purchased from Guangzhou Jiaye Food Ingredient Co., Ltd. (Guangzhou, China). Baking soda (sodium bicarbonate, NaHCO_3_) was purchased from a local supermarket. Food-grade ethanol was purchased from Shanghai Lizhi Chemical Co., Ltd. (Shanghai, China).

### 2.2. Preparation of Mashed Potatoes (MPs)

#### 2.2.1. Preparation of MPs with or without Anthocyanin

According to our previous publication [11], the ratio of water to potato flakes was set at 4:1, and a mixed hydrocolloid of 1% KG and XG (KG:XG = 3:2) was added to it. After the mixture became homogeneous, hot water was added and stirred. Then, the mixture was placed in a water bath at 70 ± 0.2 °C for 30 min to fully dissolve the hydrocolloid. During the heating process, the mixture was sealed to avoid water evaporation. Finally, the mixture was cooled to room temperature, and 0.1% anthocyanins were then added and mixed well for subsequent experiments.

#### 2.2.2. Preparation of MPs Containing Curcumin

First, 25 g of potato snowflake flour and 1% food colloid (KG and XG, with a ratio of 3:2) were mixed well; then, 100 g, 97 g, and 94 g of boiling water were added and stirred evenly. Then, the mixture was heated in a 70 ± 0.2 °C water bath for 30 min to allow the colloid to dissolve completely. During the water bath, a layer of plastic wrap was placed over the container to prevent water from evaporating. After cooling the MPs to room temperature, 0.2% baking soda was added and mixed well. Then, 0 g, 3 g, and 6 g of curcumin emulsion were added. The percentage of curcumin emulsion (preparation method is provided below) in the total liquid volume (curcumin emulsion and water) of the MPs was 0%, 3%, and 6%, respectively. The MPs were used within 1 h.

### 2.3. Preparation of Lemon Juice–Starch Gel (LJSG)

First, 20 g, 25 g, 30 g, and 35 g of potato starch were added to 100 g of concentrated lemon juice, and the mixture was preliminarily stirred by hand for about 5 min before being thoroughly mixed using a homogenizer (T18BS25, IKA Co., Chengdu, China). The mixture was then cooked at 97 ± 0.3 °C for 20 min to completely gelatinize the potato starch (with a central temperature of 95 °C or higher). The container was sealed during cooking to prevent the loss of internal moisture and the entry of external moisture. After cooking, the mixture was cooled to room temperature and used for further experiments.

### 2.4. Preparation of Curcumin Emulsion

The curcumin emulsion was prepared according to the method described by Chen et al. [10]. First, Tween 80 was dissolved in deionized water and hydrated for 4 h to obtain a water phase with a concentration of 2%. The curcumin powder (1.5 g) was dissolved in medium-chain triglycerides (MCTs) and sonicated in an ice bath at 390 W for 30 min (JYD-900L, Shanghai Zhixin Co., Shanghai, China). The mixture was then centrifuged at 3000 rpm for 2 min to remove any undissolved curcumin (2-16PK, Sigma Co., Saint Louis, MO, USA), resulting in a curcumin-containing oil phase. The water phase and oil phase were mixed in a 3:1 (*w*/*w*) ratio and homogenized using a high-speed homogenizer (JHG-54-P100, Priesen Fusion Co., Shanghai, China) for 6 min, followed by four cycles of processing using a high-pressure homogenizer at 60 MPa to obtain the final curcumin emulsion.

### 2.5. Testing the Encapsulation Effect of Curcumin in the Emulsion

Curcumin solutions of different concentrations in water and ethanol were prepared, and color tests were performed by adding an equal amount of 5% sodium bicarbonate to each solution and shaking it to observe the color change, together with the curcumin emulsion.

### 2.6. The 3D Printing Method

A two-nozzle 3D printer (SHINNOVE-D1, Shiyin Technology Co., Hangzhou City, China) was used in this study. In the stacked structure of anthocyanin-containing MPs and LJSG, both the upper and lower parts were rectangular prisms measuring 32 × 32 × 3 mm^3^, with the upper part printed using MPs and the lower part printed using LJSG. In addition, a hollow cylinder (outer diameter 30 mm, inner diameter 20 mm, and height 30 mm) was printed to evaluate the self-supporting performance of LJSG with different starch concentrations, and a flat cylinder (diameter 30 mm and height 4 mm) was printed to study the color change in MPs containing curcumin during microwave heating. The models used in the experiment were designed using Rhinoceros 5.0. Printing parameters were set following our previous publication [12]: nozzle diameter 1.2 mm, layer height 1.2 mm, printing speed 25 mm/s, and printing temperature 25 °C. Based on preliminary experiments, the relative position of the second nozzle was set to X-66.5 mm and Y-0.8 mm.

### 2.7. Rheological Properties

A rheometer was used to test the rheology (Discovery HR-2, TA Co., New Castle, DE, USA), as described in our previous publication [13], with slight modifications. A flat plate with a diameter of 20 mm was used, and the testing gap was set to 1000 µm. Before testing, excess material around the fixture was removed, and a thin layer of silicone oil was applied to prevent water evaporation during testing. The equilibration time was 5 min. In static rheological testing, the temperature was 25 °C, and the shear rate was 0.01~100 s^−1^, or testing was stopped when the sample began to rotate out of the plate. Prior to dynamic rheological testing, an amplitude sweep test was conducted in the range of 10 rad/s and 0.01~30% strain to determine the linear viscoelastic region (LVR). Then, dynamic rheological testing was performed at 25 °C, 0.1% strain (in LVR), and 1~100 rad/s. To determine the structural recovery properties, the temperature was 25 °C, the frequency was 10 rad/s, and testing was first performed at 0.1% strain (in LVR) for 120 s, followed by testing at 20% strain (not in LVR) for another 120 s, and then testing again at 0.1% strain for a final 120 s. The shear recovery properties were characterized based on the ratio of G’ at the first 30 s of the third stage to the average G’ of the first stage [13]. Each sample was tested three times.

### 2.8. Color Measurement

The color was measured with a hand-held colorimeter [10] (CR-400, Konica Minolta Co., Tokyo, Japan). Firstly, the colorimeter was calibrated with a whiteboard. During the test, the surface of the printed sample was covered with a layer of transparent plastic wrap to prevent the lens from being polluted. The lens of the colorimeter was vertically and gently attached to the surface of the sample. After printing, the 3D-printed MP–LJSG samples containing anthocyanins were treated for 1 min, 2 min, 3 min, and 4 min at a microwave power of 119 W (Glanz Co., Zhongshan, China) for color measurement. The 3D-printed MP samples containing curcumin were also treated at low microwave power (119 W) for 1 min, 2 min, and 3 min before color measurement. Two samples were selected for each measurement, and two measurements were taken randomly at different locations; the mean value was used for data analysis.

### 2.9. Low-Field Nuclear Magnetic Resonance (LF-NMR) Analysis

Water distribution was determined using an LF-NMR analyzer (Suzhou Niumag Analytical Instrument Corporation, Suzhou, China), according to Liu et al. [3]. Before testing, calibration was performed using an oil sample. About 4 g of the sample was taken for each test, wrapped in a layer of plastic wrap, and then inserted into a sample tube, which was then inserted into the LF-NMR analyzer. The parameter settings were as follows: sampling point TD = 175,002, spectral width SW = 100 kHz, echo number 3000, repeated scanning times NS = 4, and sampling repetition time TW = 3000 ms. Each sample was tested three times.

### 2.10. Data Analysis

Significance analysis was performed using one-way analysis of variance (ANOVA) and Duncan’s multiple-comparison tests in SPSS 19.0, with a significance level of *p* < 0.05. Graphs were drawn using Origin 9.0.

## 3. Results and Discussion

### 3.1. Description of This Research on pH-Induced 4D-Printed Color-Changing Food

With a dual-nozzle 3D printer, it is possible to achieve the internal nesting or stacking printing of various food materials with specific shapes. By adding inducers (acids or bases) and responsive substances (pH-sensitive pigments, such as anthocyanins or curcumin) into different parts of 3D-printed structures, automatic 4D color change in the printed samples can be achieved during microwave post-treatment. This study utilizes the principle that anthocyanins and curcumin show different colors at different pH values. Based on the dual-nozzle 3D printing technology, stacked structures with MPs (upper part, containing anthocyanins) and LJSG (lower part) and nested structures with MPs were designed and constructed to study automatic color changes in 3D-printed foods during microwave post-treatment. 

### 3.2. The 4D-printed Color-Changing Stacked Food Structures Using MP–LJSG Containing Anthocyanins Induced by pH during Microwave Post-Treatment

#### 3.2.1. The Influence of Starch Content on the Rheological Properties of Lemon Juice–Starch Gel (LJSG) System

The minimum pressure that triggers the start of fluid flow is closely related to the flow stress *τ_f_* of the fluid (Liu et al., 2019) [13]. Due to the discontinuity of the material extruded during the 3D printing extrusion stage, the ideal material should have a lower *τ_f_*. Materials with higher *τ_f_* require the system to generate greater force to maintain frequent starting or stopping of extrusion, which can easily lead to extrusion difficulties [13]. Table 1 shows that the τ*_f_* of the LJSG increase with the increase in starch content. This is obviously because the higher the starch concentration, the greater the corresponding gel strength. Figure 1A,B show the effect of different starch concentrations on the system’s viscosity (ƞ) and shear stress, indicating that the LJSG exhibits shear-thinning behavior. The shear-thinning property is conducive to the extrusion of materials during 3D printing, because the material can quickly reduce ƞ under the action of shear force to promote the smooth extrusion of the material, and then quickly recover ƞ after extrusion to promote the bonding between the printed layers [14]. The higher the starch content, the higher the ƞ and shear stress of the gel, indicating that the extrusion difficulty during 3D printing will increase. The viscosity data of the LJSG were fitted with the power law model to obtain the consistency index (*K*) and flow index (*n*) of the system.
η = *Kγ*^:^ ^(^*^n^*
^− 1)^(1)
where ƞ means viscosity, *K* means the consistency index, *γ^:^* means the shear rate, and *n* means the flow index (*n*) that describes the material’s properties of shear thinning, thickening, or whether it remains Newtonian. It can be seen from Table 1 that with the increase in starch content, the *K* of the system increased, and the flow index *n* was less than 1, further indicating that the system exhibited shear-thinning behavior. During the experiment, it was found that although the starch addition increased the extrusion difficulty of LJSG, even at a starch content of 35 g/100 g, the system still had good extrudability. This was probably because the 3D printer used in this study was sufficient enough to enable the extrusion of LJSG.

Materials with fast structural recovery characteristics are very beneficial in extrusion-based 3D printing because materials with this property will quickly restore their mechanical properties when the shear force is removed, thus maintaining the stability of the 3D printing structure (Liu et al., 2019) [13]. Figure 1C shows the shear recovery characteristics of the system under low/high/low strain at different starch concentrations. As can be seen from the figure, the material’s G′ significantly decreased under high-strain conditions (20%) but quickly recovered under subsequent low-strain conditions (0.1%). After calculation, the recovery ratios of G′ within 30 s for the system with starch concentrations of 20 g/100 g, 25 g/100 g, 30 g/100 g, and 35 g/100 g were 92.45%, 93.33%, 92.86%, and 90.98%, respectively. All of the inks illustrated good shear recovery performance, which was beneficial for the structural stability of the printed samples. Previous publications also reported that an increase in shear recoverability is beneficial for the stability of 3D-printed parts [13].

During the self-supporting stage of 3D printing, the material needs to have sufficient mechanical strength to maintain the stability of the printing structure. At this stage, the material’s G′, G*, and τ_f_ have an important influence on the structural stability [1,12]. In this experiment, the printing temperature and ambient temperature were both 25 °C, so τ_f_ can be used to characterize the degree of difficulty of initiating material flow, and both reflect the structural stability of 3D printing. G* can to some extent reflect the material’s mechanical strength. Obviously, the higher the mechanical strength, the better the supporting performance. As shown in Figure 1D–F, G′, G″, and G* of the system all had obvious frequency dependence; that is, they all increased with the increase in angular frequency. G′ was significantly higher than G″, indicating that the system exhibited more obvious solid-like characteristics. In addition, with the increase in starch content, G′, G″, and G* of the gel system all significantly increased, because the higher the starch concentration, the denser the structure formed via starch gelation, and the higher the mechanical strength. The increase in starch concentration led to an increase in the number of starch molecules per unit volume, and increased the probability of intermolecular hydrogen bonding, leading to a more compact network structure. Thus, the strength of the gel increased [15]. As shown in Figure 1H, the printed hollow cylinder (with an outer diameter and height of 30 mm) collapsed when the starch concentration was 20 g/100 g, because the mechanical strength of the material system under this formula was weak (lower G′, G*, and τ_f_), so it could not maintain the structural stability of the printing. However, although the material system under this formula could not maintain the structural stability of the hollow cylinder, it was sufficient to maintain the structural stability of the flattened cylinder (32 × 32 × 3 mm^3^).

#### 3.2.2. The Effect of Different Starch Concentrations on the Moisture State of LJSG

Since the moisture state is closely related to the rheological properties of the material and the migration of solutes between substances, the moisture state of the gel system at different starch concentrations was determined using LF-NMR. As shown in Figure 1G, the NMR spectra of LJSG have two signal peaks. With an increase in starch content, the signal peak shifts to the left, indicating that the fluidity of water gradually decreases. According to Table 2, the relaxation times corresponding to the two signal peaks were 3.18–5.54 ms (T_21_, corresponding to peak area A_21_) and 44.49–77.53 ms (T_22_, corresponding to peak area A_22_), respectively. The relaxation time of the first peak (T_21_) was below 10 ms, representing poorly flowing water, which is usually tightly bound to macromolecules such as starch and protein molecules [16]. In contrast, A_22_ accounted for a large proportion, with a longer relaxation time, representing highly fluid water, which has a greater impact on the rheological properties of the material and the migration of solutes. With an increase in starch content, the proportion of A_21_ increased, while the proportion of A_22_ decreased overall, indicating that the higher the starch content, the lower the proportion of easily flowing water in the system, which will affect the migration rate of solutes such as hydrogen ions in the system. In addition, this also indicates that the higher the starch content, the denser the structure formed, and the stronger the mechanical properties of the corresponding gel system. This can explain the phenomenon that G’, G*, and η of the system increased with increasing starch content. In a previous study, it has also been reported that the water mobility of the gel is closely related to the rheological properties and could be an indicator of G’ and G* [3].

#### 3.2.3. The influence of Different Starch Concentrations on the Automatic Color Change in the 3D-Printed Stacked Structure of MP–LJSG during Microwave Post-Treatment

The dual-nozzle 3D printing technology can create a stacked structure with multiple food materials. When there is a difference in solute concentration between two food materials, this difference will gradually disappear with the migration of water and solutes over time (or accelerated by heating). Anthocyanins are natural pigments that have antioxidant, antiaging, and anticancer properties. Anthocyanins have different molecular structures and colors at different pH levels [5]. Based on this property of anthocyanins, a stacked structure consisting of two parts was constructed in this experiment: The upper part contained MPs with anthocyanins (response part), and the lower part contained LJSG (pH stimulus layer). Concentrated lemon juice was chosen in the experiment because of its acidic property and yellow color, which is a naturally acidic food. In addition, concentrated lemon juice is often used as an ingredient in desserts and can be used to make a product similar to juice gummies after adding starch, which has a high consumer acceptance.

The 3D-printed sample achieved automatic 4D color change during microwave heating via the migration of water and solutes. The term L* denotes the lightness value, while a* is used to indicate the degree of redness. Specifically, positive a* values indicate more redness, while negative values indicate more greenness. On the other hand, b* refers to the degree of blueness, with positive values indicating a shift toward yellow and negative values indicating a shift toward blue [17]. As shown in Table 3, the a* value of the sample was positive and gradually increased with the increase in microwave treatment time, regardless of the starch content, indicating that the redness of the sample increased after microwave treatment. This is because of the ionic nature of anthocyanins whose molecule structure changes according to the pH, resulting in color change and hue variations at different pH values [15]. The b* value was negative and gradually increased, indicating that the blueness of the sample decreased. In contrast, the L* value (brightness) of the sample remained unchanged. In addition, at the same microwave treatment time, the sample with lower starch concentration changed color more quickly (Figure 2A). This is because the gel strength of the system was weaker (indicated by lower ƞ, G′, G*, and *τ_f_*; Figure 1), and the water flow was stronger (Figure 1 and Table 1), resulting in faster migration of H^+^ and faster change in the conformation of anthocyanins in the MPs with the change in pH, leading to a quicker change in the red color. This was also reported in a previous study, which revealed that a slower change in color was achieved in a stronger gel structure for 4D-printed anthocyanin-enriched foods induced by pH change [7,15]. When MPs with anthocyanins and LJSG come into contact, due to the difference in the concentration of anthocyanin molecules and H^+^ between different materials, according to Fick’s diffusion law [15], anthocyanin molecules will migrate from the MP part to the LJSG part, and H^+^ will migrate from the LJSG part to the MP part under the concentration gradient. In addition, during microwave heating, due to water evaporation, the water inside the sample will migrate from the bottom lemon juice–starch gel part to the MP part, which will cause the H^+^ dissolved in the water to further diffuse into the MP part (Figure 2B). Therefore, the dual effect of the concentration gradient and water migration will cause H^+^ to gradually migrate into the MP material layer containing anthocyanins, thus gradually lowering the pH, increasing the acidity, and causing the color of anthocyanins to turn red [15].

### 3.3. The 4D-printed Color-Changing Nested Food Structures using MPs Containing Curcumin Induced by pH during Microwave Post-Treatment

Curcumin is a natural pigment with multiple biological activities, such as antitumor and anti-inflammatory effects. Curcumin has the characteristics of an acid–alkali indicator. At pH < 1, curcumin is in a protonated state and appears red; at pH 1~7, it appears bright yellow; when pH > 7, it turns red again. The color conversion point is between pH 8 and 9 [5,18]. Sodium bicarbonate (main component NaHCO_3_) is commonly used as a leavening agent in rice noodle products. The pH of NaHCO_3_ solution is generally less than 8. In addition, NaHCO_3_ has poor thermal stability and decomposes into Na_2_CO_3_ when heated. Na_2_CO_3_ hydrolyzes to a greater extent than NaHCO_3_ and enhances the alkalinity to a pH greater than 8. In this part, the color change in curcumin at different pH values and the decomposition of HCO_3_^−^ into CO_3_^2−^ with pH improvement were considered to realize the color change in the 3D-printed nested structure of MPs containing NaHCO_3_ (the pH stimulation part) and curcumin (the response part). The automatic color change in the 3D-printed samples was achieved during post-microwave treatment.

#### 3.3.1. Analysis of the Effect of Embedding Curcumin in Emulsion

Directly adding curcumin to the MPs containing NaHCO_3_ will result in a slow color change in curcumin. In this experiment, we sought to analyze a technique in which the color of curcumin would remain unchanged before microwave heating, and the alkalinity enhancement caused by the decomposition of NaHCO_3_ after heating would cause the color change in the sample. Therefore, to avoid direct contact between curcumin and NaHCO_3_ before microwave heating, an oil-in-water emulsion of curcumin was first prepared to achieve the embedding of curcumin. Figure 3A shows the embedding effect of the curcumin emulsion. As shown, curcumin is insoluble in water. Its solubility in water is considered to be about 400 ng/mL at pH 7.4, so it is either suspended on the water’s surface or sinks to the bottom [19]. Curcumin was well dissolved in ethanol, and the solution appeared as a clear orange–yellow color. The curcumin emulsion appeared as an opaque yellow color. After adding 1 mL of NaHCO_3_ solution, the ethanol solution of curcumin turned red significantly, and the red color gradually deepened at the beginning (1~2 mL), and then remained unchanged. In contrast, the curcumin emulsion remained opaque yellow and did not change with the addition of NaHCO_3_ solution, which indicates that the oil-in-water emulsion of curcumin can to some extent avoid direct contact between the alkaline solution and curcumin in the emulsion, preventing the color change in curcumin. Therefore, in this part, the curcumin emulsion was incorporated into MPs, which contained NaHCO_3_, to achieve the automatic color change in the 3D-printed samples induced by pH change during the post-microwave heating process.

#### 3.3.2. The Effect of Different Curcumin Concentrations on the Moisture State of Mashed Potatoes (MPs)

Figure 3B and Table 4 show the effect of different added amounts of the curcumin emulsion on the water mobility of MPs. It can be seen that the NMR spectra of MPs under different formulations have three signal peaks, and the second peak (A_22_) has the largest peak area, accounting for more than 95%. The corresponding relaxation time T_22_ was 54.76–54.79 ms, which may be the result of the highly mobile water or MCTs in the curcumin emulsion in MPs. In contrast, the relaxation time corresponding to the third peak A_22_ was above 100 ms, and the peak area was less than 1%, which may be due to the small amount of free water in MPs. Overall, the addition of the curcumin emulsion did not change the relaxation time and peak area ratio of MPs.

#### 3.3.3. The Effect of the Amount of Curcumin Emulsion Added on the Color Change in 3D-Printed Nested Food Structures during Microwave Post-Treatment

Flat cylinders with a diameter of 30 mm and a height of 4 mm were printed to characterize the color change in 3D-printed MP samples containing both curcumin emulsion and NaHCO_3_ during microwave treatment. After printing, the samples were quickly treated at low microwave power for 1 min, 2 min, and 3 min, and the color difference was measured. As shown in Figure 4, when there was no microwave treatment, as the amount of the curcumin emulsion increased, the yellow color of the sample increased (b* increased gradually). This is obviously due to the yellow color of the curcumin emulsion itself. Regardless of whether the curcumin emulsion was added, the brightness and yellowness of the sample decreased with the extension of microwave treatment time; that is, the L* and b* values gradually decreased, and the greatest decrease occurred when the curcumin emulsion addition amount was 6%. For the red–green color of the sample (a*), it can be seen that the sample without the addition of curcumin emulsion showed a slightly green color (a* was negative), and microwave treatment had no significant effect on it. In comparison, for samples containing curcumin emulsion, the red color significantly increased with the extension of microwave treatment time, and the higher the curcumin emulsion content, the deeper the redness. For example, at a curcumin emulsion content of 3%, the a* value of the sample rapidly increased from −5.65 to 4.26 after 3 min of microwave treatment, while for the sample containing 6% turmeric emulsion, it increased from −7.57 to 7.63. This is mainly due to the combined effect of the following factors: Firstly, the water-in-oil structure of the curcumin emulsion is destroyed, and curcumin is released under microwave heating. The phenol hydroxyl oxygen atom in the curcumin molecule loses a proton and ionizes into a phenol oxygen anion as the pH increases. Following conversion to the anion, the ability of curcumin to donate electrons is much improved, the range of electron activity is expanded, and it is simpler to excite electrons to generate dark effects, which causes the color to shift [20]. Secondly, HCO_3_^−^ generates CO_3_^2−^ with a higher degree of hydrolysis due to the temperature increase caused by microwave heating, which increases the pH of the whole system [21], changes the conformation of curcumin molecules, and changes the color to red (Figure 4A). Chen et al. [10] also reported an increase in the a* value of 3D-printed lotus root powder gel enriched with curcumin when heating was conducted. In addition, they also reported that the turning point occurred for the color of samples, and they attributed it to the degradation of curcumin because of excessive heating. However, in our study, the color transition phenomenon did not occur, which was probably because the strong gel structure of MPs was beneficial for the stability of curcumin.

As shown in Figure 4B, a dual-nozzle 3D printer was used, with one nozzle printing MPs only containing the curcumin emulsion and the other nozzle printing MPs containing both baking soda (NaHCO_3_) and the curcumin emulsion. Before microwave heating, the addition of baking soda resulted in the weak alkalinity of the MPs, and the curcumin molecules were embedded in the water-in-oil structure. As has been reported, although curcumin is a hydrophobic molecule, the polar groups attached to the hydrophobic main chain cause it to have certain amphiphilic properties, so as to adsorb to the oil–water interface [22]. There was little or no direct contact between the alkaline substance and curcumin, so the color of the curcumin did not change, and the two parts had similar colors. After microwave heating, HCO_3_^−^ was thermally unstable and decomposed into CO_3_^2−^ with a greater degree of hydrolysis, increasing the alkalinity of the system and raising the pH (Figure 4A). At the same time, the water-in-oil structure of the curcumin emulsion was destroyed, releasing free curcumin, and the conformation changed in a stronger alkaline environment, turning the color from yellow to reddish brown (Figure 4B). As shown in Figure 4B, the initially inconspicuous smiley structure became clear after microwave treatment due to the difference in color. Based on this principle, embedded structures containing different ingredients but with similar colors (including heat-sensitive color-changing ingredients such as baking soda and curcumin) can be printed, and hidden structural information in printed food can be revealed through color changes after heating (like a coded message that requires certain processing to extract information during the war era). This would greatly enhance the enjoyment of food and leverage the personalized customization advantages of 3D printing technology.

## 4. Conclusions

This study investigated the automatic color changing of 3D-printed food containing curcumin or anthocyanins triggered by microwave post-processing. In the stacked structure of 3D-printed MP–LJSG containing anthocyanins, the color change rate was positively correlated with the diffusion of H^+^ and anthocyanin concentrations but was negatively correlated with the starch content and the gel strength of the LJSG. In the 3D-printed MPs containing curcumin emulsion, the curcumin emulsion structure was destroyed due to microwave heating, and HCO_3_^−^ was decomposed into CO_3_^2−^ with a higher hydrolysis degree, which enhanced the alkalinity, leading to conformation changes in curcumin molecules and color changes. The nested food structure designed using dual-nozzle 3D printing enables the automated presentation of hidden data based on color changes during microwave post-processing.

Although this study provides useful insights for the production of personalized 4D-printed automatic color-changing food, we only used a simplified model system and did not consider sensory evaluation. Moreover, the nutritional value of curcumin and anthocyanins and their stability were not studied. These issues need to be addressed when considering the application of curcumin or anthocyanins in 4D-printed color-changing food, which is the direction of our next research.

We emphasize that multi-nozzle 3D printing could enable the fabrication of highly attractive multi-material constructs with higher geometric complexity by controlling material distribution in a drop-on-demand way. Combined with the automatic color change triggered by a short and easy microwave process, this technology would provide consumers remarkable and unexpected sensory experience. The information obtained from this study is important for the food industry to develop a 3D healthy food product with attractive colors that provide a more satisfying experience for consumers. This would lead to more imaginative solutions in terms of personalized diets using 3D printing.

## Figures and Tables

**Figure 1 foods-12-02055-f001:**
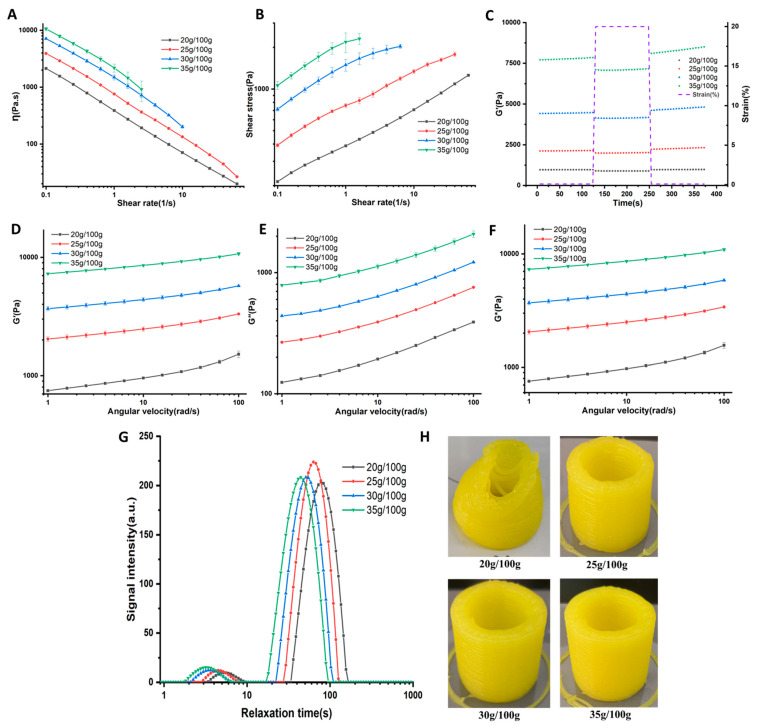
Effect of starch content on the rheological properties (**A**–**F**): (**A**) viscosity (*ƞ*); (**B**) shear stress; (**C**) shear recovery behavior; (**D**) elastic modulus (G′); (**E**) viscous modulus (G″); (**F**) complex modulus (G*)); (**G**) water mobility of lemon juice–starch gel (LJSG); (**H**) 3D-printed hollow cylinders. The reported data are the average results of the samples analyzed three times.

**Figure 2 foods-12-02055-f002:**
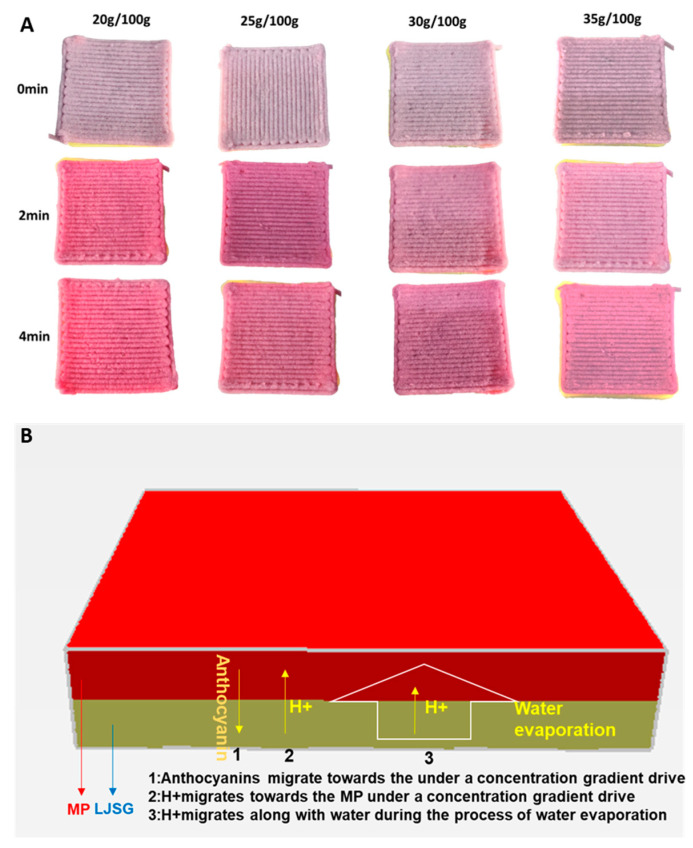
Effect of starch concentration on color change in printed samples (**A**), and solute transportation schematic (**B**) during microwave post-treatment.

**Figure 3 foods-12-02055-f003:**
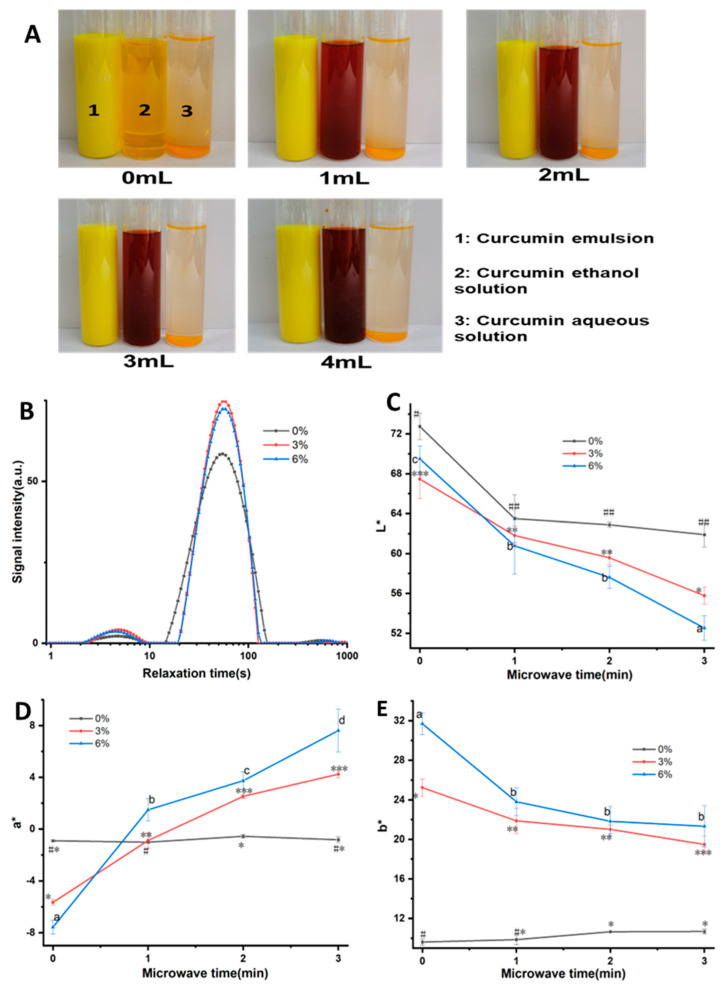
Test of embedding effect of curcumin emulsion (**A**); water mobility of mashed potatoes with different concentrations of curcumin emulsion (**B**); effect of curcumin emulsion on the color change in samples during microwave post-treatment (**C**–**E**). The different lowercase letters and “*”, “**”, “***”, “#”, “##” means the significant difference.

**Figure 4 foods-12-02055-f004:**
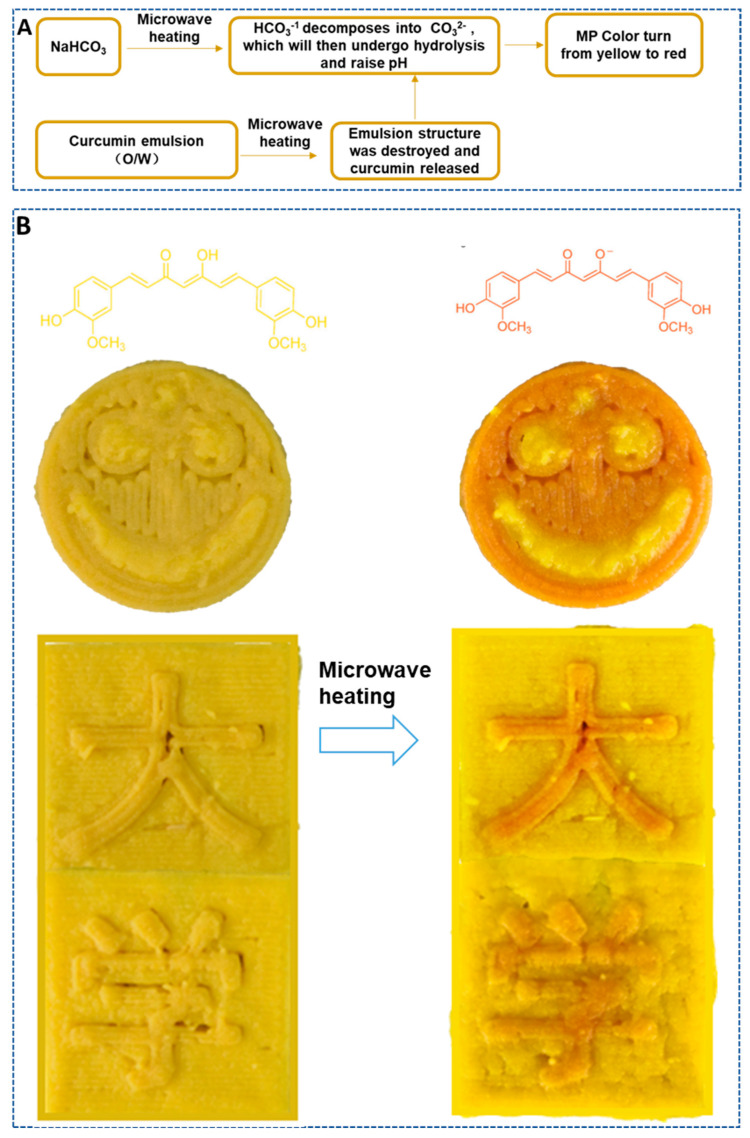
Mechanism of color change in 3D printed MPs during post-microwave treatment (**A**); microwave-triggered pH induced color change in curcumin-containing printed samples (**B**).

**Table 1 foods-12-02055-t001:** Flow stress (*τ_f_*), consistency index (*K*), and flow index (*n*) of different gel systems.

Starch Content	*τ_f_* (Pa)	Power Law Equation		
		*K* (Pa·s^n^)	*n*	R^2^
20 g/100 g	194.32 ± 13.20 ^a^	405.77 ± 6.20 ^a^	0.28 ± 0.01 ^a^	1.00
25 g/100 g	384.76 ± 36.87 ^b^	802.85 ± 14.35 ^b^	0.30 ± 0.01 ^a^	0.99
30 g/100 g	462.76 ± 40.56 ^c^	1422.71 ± 30.48 ^c^	0.30 ± 0.01 ^a^	0.98
35 g/100 g	582.22 ± 23.34 ^d^	1982.72 ± 106.54 ^d^	0.28 ± 0.03 ^a^	0.93

Note: Different letters in the same column indicate significant differences, *p* < 0.05. Each sample was tested three times.

**Table 2 foods-12-02055-t002:** Water mobility analysis of lemon juice–starch gel system with different formulations.

Starch Content	T_21_ (ms)	T_22_ (ms)	A_21_ (%)	A_22_ (%)
20 g/100 g	5.54 ± 0.05 ^a^	77.53 ± 0.00 ^a^	3.47 ± 0.10 ^a^	96.53 ± 0.10 ^a^
25 g/100 g	4.35 ± 0.21 ^b^	62.95 ± 0.00 ^b^	3.61 ± 0.19 ^a^	96.41 ± 0.17 ^a^
30 g/100 g	3.65 ± 0.00 ^c^	51.11 ± 0.00 ^c^	4.17 ± 0.01 ^b^	95.83 ± 0.00 ^b^
35 g/100 g	3.18 ± 0.00 ^d^	44.49 ± 0.00 ^d^	5.23 ± 0.11 ^c^	94.73 ± 0.17 ^c^

Note: Different letters in the same column indicate significant differences, *p* < 0.05. Each sample was tested three times.

**Table 3 foods-12-02055-t003:** Color analysis of sample during post-microwave treatment at different times.

Starch Content	Time	L*	a*	b*
20 g/100 g	0 min	27.02 ± 0.29 ^a^	6.56 ± 0.32 ^a^	−4.18 ± 0.08 ^a^
	1 min	26.70 ± 0.12 ^a^	10.94 ± 0.29 ^b^	−2.97 ± 0.25 ^b^
	2 min	26.34 ± 0.91 ^ab^	13.55 ± 0.45 ^c^	−1.54 ± 0.29 ^c^
	3 min	26.71 ± 1.66 ^ab^	15.67 ± 0.15 ^d^	−0.52 ± 0.28 ^d^
	4 min	25.04 ± 0.55 ^b^	16.46 ± 0.12 ^e^	0.17 ± 0.08 ^e^
25 g/100 g	0 min	27.11 ± 0.12 ^a^	6.34 ± 0.04 ^a^	−4.18 ± 0.03 ^a^
	1 min	26.74 ± 0.10 ^b^	9.65 ± 0.40 ^b^	−3.45 ± 0.04 ^b^
	2 min	26.46 ± 0.23 ^b^	12.26 ± 0.05 ^c^	−2.09 ± 0.14 ^c^
	3 min	26.37 ± 0.28 ^b^	14.71 ± 0.25 ^d^	−1.01 ± 0.02 ^d^
	4 min	26.11 ± 0.12 ^b^	16.29 ± 0.03 ^e^	−0.22 ± 0.11 ^e^
30 g/100 g	0 min	27.12 ± 0.26 ^a^	6.27 ± 0.11 ^a^	−4.15 ± 0.04 ^a^
	1 min	26.99 ± 0.18 ^a^	8.74 ± 0.20 ^b^	−3.83 ± 0.18 ^b^
	2 min	26.56 ± 0.19 ^b^	11.64 ± 0.39 ^c^	−2.51 ± 0.43 ^c^
	3 min	26.45 ± 0.20 ^b^	13.42 ± 0.37 ^d^	−1.33 ± 0.46 ^d^
	4 min	26.43 ± 0.40 ^b^	14.56 ± 0.32 ^e^	−0.59 ± 0.07 ^e^
35 g/100 g	0 min	27.85 ± 1.22 ^a^	6.32 ± 0.12 ^a^	−4.19 ± 0.10 ^a^
	1 min	27.67 ± 1.06 ^a^	7.95 ± 0.50 ^b^	−4.06 ± 0.04 ^a^
	2 min	26.67 ± 0.67 ^a^	10.67 ± 0.63 ^c^	−3.37 ± 0.07 ^b^
	3 min	26.55 ± 0.39 ^a^	12.43 ± 0.51 ^d^	−2.12 ± 0.06 ^c^
	4 min	26.66 ± 0.29 ^a^	13.71 ± 0.51 ^e^	−1.07 ± 0.05 ^d^

Note: Different letters in the same column indicate significant differences for each formulation, *p* < 0.05. Each sample was tested four times.

**Table 4 foods-12-02055-t004:** Water mobility analysis of mashed potatoes with different concentrations of curcumin emulsion.

Curcumin Emulsion	T_21_ (ms)	T_22_ (ms)	T_23_ (ms)	A_21_ (%)	A_22_ (%)	A_23_ (%)
0%	5.00 ± 0.25 ^a^	54.79 ± 0.00 ^a^	541.59 ± 0.00 ^a^	2.84 ± 0.30 ^a^	96.45 ± 0.41 ^a^	0.71 ± 0.12 ^c^
3%	4.66 ± 0.23 ^ab^	54.79 ± 0.00 ^a^	943.79 ± 0.00 ^c^	3.89 ± 0.20 ^b^	95.91 ± 0.21 ^a^	0.20 ± 0.01 ^a^
6%	4.35 ± 0.21 ^b^	56.76 ± 2.79 ^a^	690.97 ± 33.91 ^b^	3.72 ± 0.05 ^b^	95.94 ± 0.07 ^a^	0.34 ± 0.03 ^b^

Note: Different letters in the same column indicate significant differences, *p* < 0.05. Each sample was tested three times.

## Data Availability

Data is contained within the article.

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
