# Peer review of "Microwave-Triggered 4D Automatic Color Change in 3D-Printed Food Materials Incorporating Natural Pigments"

_foods, 2023, doi:10.3390/foods12102055_

Round 1

Reviewer 1 Report

Manuscript ID: foods-2338732

Manuscript Title Microwave-triggered 4D spontaneous color change of 3D printed food materials incorporating natural pigment

Comments: The authors present a very interesting manuscript about the the use of microwave to stimulate color change of 3D printed food containing natural pigments. The subject is interesting and the results are relatively well interpreted, however, the manuscript organization is confusing and makes reading it a little hard.

Some detailed comments are appended in the pdf file

Author Response

Reviewer 1:

Comments: The authors present a very interesting manuscript about the the use of microwave to stimulate color change of 3D printed food containing natural pigments. The subject is interesting and the results are relatively well interpreted, however, the manuscript organization is confusing and makes reading it a little hard. Some detailed comments are appended in the pdf file.

Response: Thanks very much for your general positive comments. As suggested, several issues have been addressed. Detailed revisions have been marked in blue in the revised manuscript. The manuscript organization has been emphasized by using different title forms and numbering. Detaild revisions please see the revised manuscript marked in blue. We believe the revisions made accordingly would improve the manuscript. Thank you.  

Reviewer 2 Report

The paper has some significant flaws in its methodology, presentation of results, and discussion of findings. The authors need to revise and improve the materials and methods, results, and conclusion sections to ensure that they are comprehensive, accurate, and well-supported by previous research.

The abstract summarizes the study's purpose, methods, results, and conclusions clearly and concisely. However, it lacks context and broader implications of the findings. The introduction presents the problem and importance of the research in 4D food printing. However, it lacks background information, a comprehensive literature review, and specific research questions or hypotheses.

There is a significant issue with the materials and methods section, as the authors did not cite any previously published methods used in their research. They should review and revise this section to include appropriate references to previous works that influenced their methodology.

Several issues need to be addressed in the results section, including concerns about statistical analysis, unclear table titles, and limited presentation of color values. The results section should be revised to address these issues and ensure that the data is presented in a clear and accurate manner. The discussion section should provide a critical analysis of the results, relate them to previous research in the field, and explain why the results are the way they are.

The conclusion section could benefit from a more explicit statement about the significance of the findings and their potential impact on the food industry or related fields. Additionally, some limitations of the study could be discussed in more detail, such as the use of a simplified model system and the lack of sensory evaluation.

Author Response

Response to Reviewer 2 Comments

Reviewer 2:

The paper has some significant flaws in its methodology, presentation of results, and discussion of findings. The authors need to revise and improve the materials and methods, results, and conclusion sections to ensure that they are comprehensive, accurate, and well-supported by previous research.

Response: Thanks very much for your suggestions. I have revised the manuscript considerably as suggested, which has been marked in blue in the revised manuscript. Thank you, and we sincerely hope this has improved the manuscript.

The abstract summarizes the study's purpose, methods, results, and conclusions clearly and concisely. However, it lacks context and broader implications of the findings.

Response: Thanks very much for your suggestions. We have revised the abstract and give the possible implications as follows: “This study indicated that 4D printing could realize the creation of colorful and attractive food structures, and bring us more imagination about personalized foods, which can be particularly important to people with poor appetite”. Thanks very much and we sincerely hope this would make sense. Thank you.

The introduction presents the problem and importance of the research in 4D food printing. However, it lacks background information, a comprehensive literature review, and specific research questions or hypotheses.

Response: Thanks very much for your suggestions. We have added and revised several parts of the introduction: “However, the number of literature available for color-changing 4D food printing is scanty. Ghazal et al. (2019) and He et al. (2020) utilized the principle of color change in anthocyanins at different pH levels, by spraying acidic solutions on the surface of printed samples or printing materials containing anthocyanins with different acid-base materials, but in the above studies the significant color changes often required several hours or even longer, which hindering its application. It is highly required that a rapid color change of 3D printed foods could be achieved to realize the broader implication”. “To our knowlede, there are only two publications utilized microwave as heating source to achieve a quick spontaneous color change of 3D printed structures. Guo et al. (2021) used the microwave as the stimulus source and gelatin-gum arabic complex coacervates as the stimulus material to realize the color and flavor changes of the printed dough. Chen et al. (2021a) using microwave to stimulate the color change of curcumin lotus root gel printed by 3D. However, in their studies only one kind of food material was used to construct food structures, which restricted the development of multi ingredient and multi color foods. In our study, using dual-nozzle 3D printing technology, food structures containing both responsive parts (with pH-sensitive plant pigments) and stimulating parts (with acid or alkali) were constructed to achieve spontaneous color changes during post-processing. To realize the controlled deposition of material and the creation of highly attractive multi-material constructs with higher geometric complexity and appealing appearance, a dual-nozzle printer was used in this study to create the 4D printed color-changing foods”. Thanks very much and we sincerely hope this would make sense.  

There is a significant issue with the materials and methods section, as the authors did not cite any previously published methods used in their research. They should review and revise this section to include appropriate references to previous works that influenced their methodology.

Response: Thanks very much for your suggestions. In materials and methods part, references have been added for each part, and relevant revisions have been marked in blue. Thank you.

Several issues need to be addressed in the results section, including concerns about statistical analysis, unclear table titles, and limited presentation of color values. The results section should be revised to address these issues and ensure that the data is presented in a clear and accurate manner. The discussion section should provide a critical analysis of the results, relate them to previous research in the field, and explain why the results are the way they are.

Response: Thanks for your suggestions. As suggested, we have carefully revised the manuscript in many places, such as changing the presentation of color values in Table 3, relating the results to previous research, giving possible explaination, adding more references. Detailed information please see the revised manuscript marked in blue. Here, only part of revisions were listed as follows.  

Color values:

Table 3. Color analysis of sample during post-microwave treatment at different time

Starch content

Time

L*

a*

b*

20g/100g

0 min

27.02±0.29a

6.56±0.32a

-4.18±0.08a

1 min

26.70±0.12a

10.94±0.29b

-2.97±0.25b

2 min

26.34±0.91ab

13.55±0.45c

-1.54±0.29c

3 min

26.71±1.66ab

15.67±0.15d

-0.52±0.28d

4 min

25.04±0.55b

16.46±0.12e

0.17±0.08e

25g/100g

0 min

27.11±0.12a

6.34±0.04a

-4.18±0.03a

1 min

26.74±0.10b

9.65±0.40b

-3.45±0.04b

2 min

26.46±0.23b

12.26±0.05c

-2.09±0.14c

3 min

26.37±0.28b

14.71±0.25d

-1.01±0.02d

4 min

26.11±0.12b

16.29±0.03e

-0.22±0.11e

30g/100g

0 min

27.12±0.26a

6.27±0.11a

-4.15±0.04a

1 min

26.99±0.18a

8.74±0.20b

-3.83±0.18b

2 min

26.56±0.19b

11.64±0.39c

-2.51±0.43c

3 min

26.45±0.20b

13.42±0.37d

-1.33±0.46d

4 min

26.43±0.40b

14.56±0.32e

-0.59±0.07e

35g/100g

0 min

27.85±1.22a

6.32±0.12a

-4.19±0.10a

1 min

27.67±1.06a

7.95±0.50b

-4.06±0.04a

2 min

26.67±0.67a

10.67±0.63c

-3.37±0.07b

3 min

26.55±0.39a

12.43±0.51d

-2.12±0.06c

4 min

26.66±0.29a

13.71±0.51e

-1.07±0.05d

 Note:Different letters in the same column indicate significant difference for each formulation, p<0.05. Each data was tested four times.

       On the other hand, b* refers to the degree of blueness, with positive values indicating a shift towards yellow and negative values indicating a shift towards blue (Messina et al., 2012). As shown in Table 3, with the extension of microwave treatment time, the a* value of the sample is positive and gradually increases with time, regardingless of the starch content, indicating that the redness of the sample increases after microwave treatment. This was because that the ionic nature of anthocyanins whose molecule structure changes according to the pH, resulting in color change and hues variations at different pH values (Ghazal et al., 2021).

Disscussion:

Firstly, the water-in-oil structure of curcumin emulsion is destroyed and curcumin is released under microwave heating. The phenol hydroxyl oxygen atom in the curcumin molecule lost a proton and ionized into phenol oxygen anion as the pH rised. Following conversion to anion, the ability of curcumin to donate electrons was much improved, the range of electron activity was expanded, and it was simpler to excite electrons to generate dark effects, which caused color to shift (Etxabide et al., 2021). Secondly, HCO3- generates CO32- with a higher degree of hydrolysis due to the temperature increase caused by microwave heating, which increases the pH of the whole system (Cai et al., 2022), changes the conformation of curcumin molecules, and changes the color to red (Figure 4A). Chen et al. (2021) also reported that the increase of a* value of 3D printed lotus root powder gel enriched with curcumin when heating was conducted. In addition, they also reported that the turning point occurred for the color of samples and they attributed it to the degradation of curcumin because of excessive heating. However, in our study, the color transition phenomenon did not occur, which wa probably because the strong gel structure of MP was helpful for the stability of curcumin.      

References:

Kharat, M., Du, Z., Zhang, G., & McClements, D. J. (2017). Physical and chemical stability of curcumin in aqueous solutions and emulsions: Impact of pH, temperature, and molecular environment. Journal of Agricultural and Food Chemistry, 65(8), 1525–1532.

Tønnesen, H. H., Masson, ´ M., & Loftsson, T. (2002). Studies of curcumin and curcuminoids. XXVII. Cyclodextrin complexa-tion: Solubility, chemical and photochemical stability. International Journal of Pharmaceutics, 244(1), 127–135

Etxabide, A., Kilmartin, P.A., Mat´ e, J.I., 2021. Color stability and pH-indicator ability of curcumin, anthocyanin and betanin containing colorants under different storage conditions for intelligent packaging development. Food Control 121, 107645.

Ghazal, A.F., Zhang, M., Bhandari, B., & Chen, H. (2021). Investigation on spontaneous 4D changes in color and flavor of healthy 3D printed food materials over time in response to external or internal pH stimulus. Food research international 142, 110215. https://doi.org/ https://doi.org/10.1016/j.foodres.2021.110215.

Ghazal, A.F., Zhang, M., Liu, Z.J.F., & Technology, B. (2019). Spontaneous color change of 3D printed healthy food product over time after printing as a novel application for 4D food printing.  12, 1627-1645.

Thanks very much for your valuable suggestions. Detailed revisions please see the revised manuscript.

The conclusion section could benefit from a more explicit statement about the significance of the findings and their potential impact on the food industry or related fields. Additionally, some limitations of the study could be discussed in more detail, such as the use of a simplified model system and the lack of sensory evaluation.

Response: Thanks very much for your suggestions. As suggested, we have revised the conclusions as following: “This study investigated the spontaneous color-changing of 3D printed food containing curcumin or anthocyanins triggered by microwave post-processing. In the stacked structure of 3D printed MP-LJSG containing anthocyanins, the color change rate was positively correlated with the diffusion of H+ and anthocyanin concentration, but was negatively correlated with the starch content and gel strength of the LJSG. In the 3D printed MP containing curcumin emulsion, the curcumin emulsion structure was destroyed by microwave heating, and HCO3- was decomposed into CO32- with higher hydrolysis degree, enhancing the alkalinity, leading to conformation changes of curcumin molecules and color changes. The nested food structure designed by the dual-nozzle 3D printing achieved the automated presentation of hidden information based on color changes during microwave post-processing.

Although this study provides useful information for the production of personalized 4D printed spontaneous color-changing food, in this study we only used a simplified model system and did not consider the sensory evaluation. Moreover, the nutritional value of curcumin and anthocyanins and their stability were not studied. These issues need to be addressed when considering the application of curcumin or anthocyanins in 4D printed color-changing food, which is the direction of our next research.

Anyway, we emphasize that multi-nozzle 3D printing could enable the fabrication of highly attractive multi-material constructs with higher geometric complexity by controlling the materials distribution in a drop-on-demand way. Combining with the spontaneous color-change riggered by a short and easy microwave process, this technology would provide onsumers amazing and superisng sensory experience. The information obtained from this study is important for the food industry to fabricate a 3D healthy food product with attractive colors that satisfy consumer acceptance. This would bring us more imagination about 3D printing personalized diet.” Thanks again for your suggestions.

Round 2

Reviewer 1 Report

Dear authors.

I think there are a mistake in your file. In the "author response" file, I don´t found any reply of the questions I did, just the modified manuscript.

I appreciate if the authors send me a detailed answers.

Author Response

Response to Reviewer 1 Comments

Comments: The authors present a very interesting manuscript about the the use of microwave to stimulate color change of 3D printed food containing natural pigments. The subject is interesting and the results are relatively well interpreted, however, the manuscript organization is confusing and makes reading it a little hard. Some detailed comments are appended in the pdf file.

Response: Thanks very much for your general positive comments. Detailed revisions have been marked in blue in the revised manuscript, and listed as follow point by point. The manuscript organization has been emphasized by using different title forms and numbering. Detaild revisions please see the revised manuscript marked in blue. We believe the revisions made accordingly would improve the manuscript. Thank you.  

Point to point responses are listed as followed

Nested structures?

Response: Thanks very much for your questions. Nested structures means that the food structures were printed using a two-nozzle printer by extruding the corresponding materials respectively based on the pre-designed model, such as the “smile structure” presented in the manuscript. Thanks again, and we sincerely hope this would make sense to you.

(Guo et al., 2021) should be deleted. Using should be changed as used.

Response: Thanks very much for your suggestions. As suggested, (Guo et al., 2021) has been deleted, and using has be changed as used. Thanks again.

This part “Chen et al. (2021a)………used in their experiment” should be updated with latest information.

Response: Thanks very much for your suggestions. As suggested, this part has been revised as follows: “To our knowlede, there are few publications utilized microwave as a heating source to achieve a quick automatic color change of 3D printed structures. Guo et al. (2021) used the microwave as the stimulus source and gelatin-gum arabic complex coacervates as the stimulus material to realize the color and flavor changes of the printed dough. Chen et al. (2021a) using microwave to stimulate the color change of curcumin lotus root gel printed by 3D. However, in their studies only one kind of food material was used to construct food structures, which restricted the development of multi ingredient and multi color foods. In our study, using dual-nozzle 3D printing technology, food structures containing both responsive parts (with pH-sensitive plant pigments) and stimulating parts (with acid or alkali) were constructed to achieve automatic color changes during post-processing. To realize the creation of highly attractive mul-ti-material constructs with higher geometric complexity and appealing appearance, a dual-nozzle printer was used in this study to create the 4D printed color-changing foods”. Thanks again, and we sincerely hope this would make sense to you.

should be more specific with the “spontaneous” term.

Response: Thanks very much for your suggestions. As suggested, spontaneous has been changed as automatic in the revised manuscript. Thanks again.  

“25% blueberry anthocyanin” what’s the mean? How about the purity? Is it in the powder form? 95% purity curcumin in powder?

Response: Thanks very much for your questions. We used the blueberry crude extract powder which contains 25% anthoyanin. “95% purity curcumin” also means the extract powder contains 95% curcumin. Thanks again, and we sincerely hope this would make sense to you.

Improve the sentence “During the heating process, a layer of plastic wrap is covered on the surface of the container to avoid water evaporation”.

Response: Thanks very much for your suggestions. The sentence has been changed as “During the heating process, the mixture is sealed to avoid water evaporation.” Thanks again.

Improve the sentence “Based on this, by adding inducers (acids or bases) and responsive substances (pH-sensitive pigments, anthocyanins, and curcumin) into different food materials, and combining the customized advantages of the dual-nozzle 3D printing technology with multiple material structures, spontaneous 4D color change of the printed samples can be achieved during microwave post-treatment.”

Response: Thanks very much for your suggestions. The sentence has been changed as “By adding inducers (acids or bases) and responsive substances (pH-sensitive pigments, such as anthocyanins or curcumin) into different parts of 3D printed structures, auto-matic 4D color change of the printed samples can be achieved during microwave post-treatment.” Thanks again, and we sincerely hope this would make sense to you.

Improve the sentence “The higher the starch content, the higher the Æž and shear stress of the gel system under the same shear rate, which means that the extrusion smoothness of the system during 3D printing will decrease as the starch content increases”.

Response: Thanks very much for your suggestions. The sentence has been changed as “The higher the starch content, the higher the Æž and shear stress of the gel, indicating that the extrusion difficulty during 3D printing will increase.” Thanks again.

“spontaneous color change” means naturally? By heat induction?

Response: Thanks very much for your questions. “spontaneous” has been changed as “naturally”, which means that the color of samples could change automatically during microwave post-processing.

Which pH values we are talking about? Why the authors used acid system?

Response: Thanks very much for your questions. The pH value of lemon juice gel was around 3~4. The acid system was used because anthocyanin molecules will experience a color change when interacts with H+. Moreover, concentrated lemon juice is often used as an ingredient in desserts and can be used to make a product similar to juice gummies after adding starch, which has a high consumer acceptance. Thanks again and we sincerely hope this would make sense to you.

In the lines “indicating the redness of samples……. the blueness of the sample decreases”, the redness of the system is stable in this Ph? The blueness was because of the acid system?

Response: Thanks very much for your questions. The redness was not stable initially and increased with time, but it gradually became stable with the microwave post-processed time. The changes of blueness and redness of samples was because that the ionic nature of anthocyanins whose molecule structure changes according to the pH, resulting in color change and hues variations at different pH values. Thanks again and we sincerely hope this would make sense to you.

This study studied, please change.

Response: Thanks very much for your suggestions. This sentence has been changed as “This study investigated the automatic color-changing of 3D printed food containing….”. Thanks again.

There are few references.

Response: Thanks very much for your suggestions. Five more references have been added as suggested as follows.

Tønnesen, H. H., Masson, ´ M., & Loftsson, T. (2002). Studies of curcumin and curcuminoids. XXVII. Cyclodextrin complexa-tion: Solubility, chemical and photochemical stability. International Journal of Pharmaceutics, 244(1), 127–135

Kharat, M., Du, Z., Zhang, G., & McClements, D. J. (2017). Physical and chemical stability of curcumin in aqueous solutions and emulsions: Impact of pH, temperature, and molecular environment. Journal of Agricultural and Food Chemistry, 65(8), 1525–1532.

Etxabide, A., Kilmartin, P.A., Mat´ e, J.I., (2021). Color stability and pH-indicator ability of curcumin, anthocyanin and betanin containing colorants under different storage conditions for intelligent packaging development. Food Control, 121, 107645.

Ghazal, A.F., Zhang, M., Bhandari, B., & Chen, H. (2021). Investigation on spontaneous 4D changes in color and flavor of healthy 3D printed food materials over time in response to external or internal pH stimulus. Food research international, 142, 110215.

Ghazal, A. F., Zhang, M., & Liu, Z. (2019). Spontaneous color change of 3D printed healthy food product over time after printing as a novel application for 4D food printing. Food and Bioprocess Technology, 12, 1627-1645.

Thanks again.  

Reviewer 2 Report

 The revised version now includes all of the suggestions that I provided as a reviewer. I appreciate the effort and dedication you put into revising your manuscript, and I believe that your changes have significantly strengthened your work. The revised manuscript is more coherent, more well-supported, and more engaging than the previous version. I am confident that your article will make a valuable contribution to the field, and I am pleased to recommend it for publication. Thank you again for your hard work, and I look forward to seeing your work in print.

Author Response

Thanks very much for your positive comments, and I am very appreciate your valuable suggestions.